# Insights into Two Novel Orthopalladated Chromophores with Antimicrobial Activity against *Escherichia coli*

**DOI:** 10.3390/molecules27186060

**Published:** 2022-09-16

**Authors:** Rosita Diana, Francesco Silvio Gentile, Antonio Carella, Luigi Di Costanzo, Barbara Panunzi

**Affiliations:** 1Department of Agricultural Sciences, University of Naples Federico II, Via Università, 100, 80055 Portici, Italy; 2Department of Pharmacy, University of Salerno, Via Giovanni Paolo II, 132, 84084 Fisciano, Italy; 3Department of Chemical Sciences, University of Napoli Federico II, Strada Comunale Cinthia, 26, 80126 Napoli, Italy

**Keywords:** orthopalladated, chromophores, antimicrobia, DFT analyses

## Abstract

Advanced chromophoric tools, besides being biologically active, need to meet the expectations of the technological demands including stability, colour retention, and proper solubility for their target. Many coordination compounds of conjugated ligands are antibacterial dyes, able to combine a strong dyeing performance with a useful biological activity. Specifically, palladium (II) complexes of Schiff base ligands are known for their relevant activity against common bacteria. In this article, we report the synthesis and comprehensive experimental and theoretical characterization of two novel Pd(II) chromophore complexes obtained from a cyclopalladated Schiff base as two different chelating azo dyes. The antibacterial response of these two novel complexes was tested against the ubiquitous *Escherichia coli* bacterium in an aqueous medium and revealed a noteworthy antimicrobial activity, higher than when compared with their uncoordinated biologically active ligands.

## 1. Introduction

Antibacterial colorants and dyes are relevant topic to the researchers. Natural colorants extracted from plants, with antimicrobial properties, have been widely used both as herbal medicine and dyes for at least 4000 years for tattoos, hair dyeing, home care and remedies, and for textiles [1]. The use of antimicrobial dyes on textile fabrics for clothing protecting started in 1941 for military purposes [2], until the development of synthetic dyes in the last century reduced their use [3]. Nowadays, synthetic dyes are ubiquitous, from the textile industry to many electro-optical applications, from bioengineering to biomedical field [3,4,5,6,7,8,9,10,11,12,13].

Advanced chromophoric tools are designed to meet specific requirements, such as a high molar extinction coefficient, stability, solubility, affordability, optical response, and targeted biological activity [14]. Organic molecules often show excellent dying performance but can suffer from instability. In fact, if the presence of a double bond guarantees the desired dying effects, on the other hand, the double bond itself causes chemical, thermal, or photochemical lability. Organic molecules containing the characteristic azo double bond are considered fascinating motifs in organic chemistry. Since the last century, they have been widely employed as dyes for printing, paper, textiles, cosmetics, electronics, optics, lasers, biomedical and material sciences, etc. [15]. Their chemical and/or photo-degradation and photoisomerization are desirable in some areas, such as wastewater treatment and disposal, [16] and in sensing [17,18]. In other areas, such as in long-lasting colouring, their instability represents a disadvantage. Furthermore, the synthetic azo dyes employed for staining purposes are required to be stable and able to retain their colour. In addition, the diazo compounds are part of the photochromic family, largely used as antibacterial, antiviral, antifungal, and cytotoxic agents [19,20,21,22,23,24].

Organic molecules with the C = N double bond, also known as Schiff bases, are widely used as pigments and dyes, catalysts [25], sensors [26,27,28,29,30], optical tools, intermediates in organic synthesis [31,32], and are biologically performant. Specifically, they show antibacterial [33], antimalarial, antiproliferative [10], analgesic, anti-inflammatory, antiviral, antipyretic, and antifungal [12] properties. Interestingly, the coordination of Schiff bases-containing ligands to metals generally leads to enhanced antibacterial and antifungal effects [1,34,35,36,37,38].

Chromophore metal complexes show interesting optical properties responsible for various technologic and industrial applications, including dyes for specific substrates (textiles, liquid-crystals, bioplastics [39,40], layers for Dye-Sensitized Solar Cells (DSSC), active Non-Linear Optics (NLO) materials for ultrafast switches, lasers, and sensors [37,41,42,43,44,45,46,47,48,49,50,51]. On the other hand, the major classes of pharmaceutical agents contain examples of metal compounds in current clinical use [52]. Metal centres can organize surrounding atoms achieving pharmacophore-highly specific geometries [53]. The role of coordination compounds as biological agents and/or tools is fully recognised [1,54,55,56,57,58,59,60] and, in some cases, they can integrate with the role of dye.

In previous work, we performed a systematic investigation of orthopalladated push–pull aromatic imines as stable colorants for applications in second-order nonlinear optics (NLO) [61,62,63]. Our prior studies and experience with targeted organometallic compounds have led us, with this report, to the design, synthesis, and photophysical characterisation of two novel palladium-based chromophores, named Pd1 and Pd2 (see Figure 1). The metal ion is cyclopalladated to the Schiff base ligand and the *O, N* chelated to a diazo scaffold known for its antimicrobial properties [1,64,65]. Upon the coordination, both ligands are forced into a planar arrangement which, in turn, amplifies the conjugation pattern of the single uncoordinated moieties.

The two novel organometallic species were found to be stable and very intense chromophores, displaying different shades of orange/red. The single crystal structure of the Pd1 was analysed to deepen the structural features related to the absorbance pattern and was used as the starting point for the quantum mechanical characterization. The Density Functional Theory (DFT) investigation of the differential electron density describes the metal as an active electron bridge in the excitation process, and its electronic charge transfer was compared with the pattern of the uncoordinated ligands, drawing interesting chemical insights. Finally, their antibacterial response was tested in an aqueous medium against the ubiquitous *Escherichia coli* bacteria, and the results were compared.

## 2. Results and Discussion

### 2.1. Chemistry and Optical Response

The development of new metal complexes with optical and biological properties needs to meet the expectations of the technological demands. Dyes that are very intense, stable, and possibly soluble for a given target can be achieved by the assembly of suitable organic ligands through coordination to a metal cation. The use of metals as bridges along the conjugated organic system [66] must be compatible with a more efficient electron π-conjugation path across the metal atom. Specifically, chelating atoms within a rigid cycle favour the push–pull electronic effect of the ligands, and the metal can play a pivotal role in the charge-transfer transitions of the organic part. Square-planar coordinated metals (with dp–pp interactions) are good candidates [67]. For the aim of the present study, we hypothesised that the electron charge-transfer within the Pd1 complex would involve the whole molecule, including both the organic ligands and the metal cation. Therefore, we also expected significant differences between the single ligands and their metal complexes from a spectroscopic and even biological point of view.

Here, we used the organometallic bound [68,69,70,71,72] to design stable and non-symmetric chromophore complexes with two different organic moieties coordinated to the metal centre. The Schiff base ligand L (see Figure 1) undergoes cyclopalladation, resulting in a dinuclear specie (PdL)_2_. By a bridge-splitting reaction of the dinuclear complex with the *O, N* chelating azo ligands L1 or L2, we obtained two complexes, named Pd1 and Pd2, respectively. Locked in the cyclopalladated core, the double bond of both ligands benefits from stabilisation and is prevented from isomerisation [63,73]. In our case, the chemical stability was checked by recording the ^1^H-NMR spectra of both the Pd1 and Pd2 before and after 10 days (the samples were kept at room temperature in the condition of daylight). As expected, the NMR pattern resulted unaltered.

**Figure 1 molecules-27-06060-f001:**
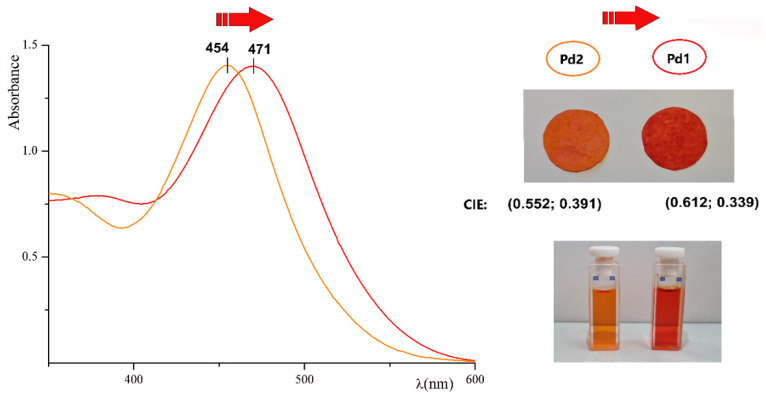
Absorbance curves of Pd1 and Pd2 complexes with concentration of 0.05 mM in acetone. Spectral red-shift from Pd2 to Pd1 is indicated by a red arrow. In the picture above: filter paper disks soaked with Pd1 (on the **right**) and Pd2 (on the **left**), with concentration of 5 mM in acetone, show a marked colour difference. The CIE colour space coordinates are reported. In the picture below: solutions of Pd1 (on the **right**) and Pd2 (on the **left**) used to record the spectra.

All ligands have a π-electron conjugated system and display various shades of an orange-yellow colour. As for many push-pull chromophores, ligand L has dying properties [73,74] and antimicrobial activity [75]. Ligand L1 belongs to a family of azo compounds, well known for their antimicrobial activity [20]. The L2 ligand was derived from the L1 by the addition of a charged trimethylpentane-ammonium chain (Figure 1). The addition of the ammonium-charged atom group in L2 was guided by two criteria, firstly, to test the antibacterial efficiency of the trimethylammonium functional group compared with the OH group [20,38]; and secondly, to improve the solubility of the complex Pd2 in aqueous solutions, desirable for the antimicrobial response in an aqueous environment [76].

The Pd1 and Pd2 complexes are red and orange crystalline solids, respectively. The R substituent (Figure 1) drives the difference in the absorbance maximum and colour. The Pd1 colour is redder than the Pd2, and a 17 nm red-shift of the absorption maxima is recorded in acetone, with a substantial difference in the CIE coordinates (see Figure 1). In agreement, their dying ability tested on filter paper substrates is also observed. Figure 1 shows the dry filter paper disks soaked with a concentered acetone solution of the Pd1 and Pd2 samples, respectively. The molar extinction coefficient, measured in acetone, at the absorption maximum has a magnitude of 10^6^. The value is relevant (about 100 times greater than the unbonded ligands [20,73]). Interestingly, it was ascertained as the result of the contribution of the metal in the charge-transfer process (Section 3.3).

Both complexes show good solubility in ordinary organic solvents, where they produce a remarkable solvatochromic effect, depending on the solvent polarity. Interestingly, as for the precursor L [77], this is an indication of the potential NLO activity of the chromophores [73]. The red-shift detected in the absorbance maximum with the increasing solvent polarity is summarized in Table 1. The absorbance pattern of the solutions used in the solvatochromic test remain unaltered for months under daylight at room temperature. Similarly, the soaked filter papers do not fade under exposure to daylight for over two months. After the same time, the absorbance spectra recorded on solid samples of the Pd1 and Pd2 deposited on quartz slides (see absorbance spectra in Appendix A and absorbance maxima in Table 1) retain the same pattern.

### 2.2. Crystal Structure of Pd1

The X-ray structure of the Pd1, characterized by the Pd(II) ion coordination of the two aromatic Schiff bases, is shown in Figure 2. The probe crystallizes in the P2_1_/c space group, with one complex in the asymmetric unit. A partially occupied water molecule or acetic acid was also detected. One of the two coordinating ligands is characterized by electron-donor groups (i.e., methoxy and dimethylamino), while the second ligand contains a hydrazone bond between two substituted aromatic groups.

The square-planar coordination around the Pd atom group is characterized by two coordinating *sp^2^* N atom groups, one from each ligand and an electron donating to the central metal, one -O atom group from the phenate ring, and finally, a C atom group from the phenyl ring, forming a metal-organic bond. The Pd atom group is within two cycles: a five-membered ring, essentially planar, and a bended six-membered salicylaldiminate ring, which in turn includes a hydrazone bond. As a consequence of the coordination to the Pd(II) ion, both the benzo-*ter*-butoxy and the *2,4*-dihydroxydrobenzene are slightly rotated with respect to each other, and with respect to the plane of the hydrazone (Figure 2). A similar consideration can be drawn for the second ligand. The structure is stabilised by the hydrogen bond interactions between a mutually exclusive water molecule or acetic acid molecule and the -OH atom group of the Pd1 ligand (Table 2). A selection of the bond angles and distances around the Pd(II) for the Pd1 is shown in Table 3.

The crystal packing of the Pd1 complex shows an anti-parallel arrangement of the ligands and exhibits van der Waals interactions of the dimethylamino benzene group along the *b*-axis, in agreement with the symmetry elements of the P2_1_/c space group. A selection of the symmetric short distance -CH_3_-π and π-π-displaced stacking interactions are shown in Appendix A. The crystallographic data statistics and structure refinement details for the Pd1 complex are reported in Table 2.

### 2.3. Theoretical Analysis

The electronic charge-transfer involved in the excitation process was examined considering the differential electron density of the complex Pd1 compared with its ligands. The differential electron density contribution is defined as ρ_diff_ = ρ_1exc_ − ρ_gr_, i.e., the difference between the electron density of the first excited state and the ground state. The analysis can be considered more representative compared with the classical orbital representation, where the electron charge transfers involve different molecular topological sites. The density plot reported in Figure 3 shows a significant difference in the electronic transfer process occurring in the Pd1 (Figure 3B) compared with the uncoordinated ligands L and L1 (Figure 3A). The L1 ligand was considered both in its neutral form and in the negative phenate form L1′ (i.e., in the chelating deprotonate form).

Due to the expected push-pull pattern, the main electron transfer contribution in the HOMO-LUMO excitation of the L can be ascribed to the migration path from the amino-substituted ring (negative zone, red lobes) to the nitrobenzene group (positive zone, blue lobes). On the opposite side, the excitation process in both the neutral and deprotonated L1 involves the aromatic diazo core with a scarce charge separation.

Interestingly, the Pd1 complex shows a different behaviour and peculiar situation: a strong electron charge transfer involving both precursors and the metal atoms was detected. Specifically, the ligand L1 transfers its electron density (red lobes) to the nitrobenzene side (blue lobes) of the other ligand L. The palladium cation plays a role in the charge-transfer process as a conductive bridge, with its own negative contribution shaped like a dz^2^ orbital (see Figure 3B). The HOMO-LUMO energy gaps of the ligands are comparable (respectively 2.46, 2.50, 2.93 eV for L, L1 and L1′), lower for Pd1 (2.20 eV).

### 2.4. Antibacterial Response

Many coordination compounds with a high molar extinction coefficient are known for both their biological activity and dyeing performance [1,14]. The palladium (II) complexes with Schiff base ligands are known, as well, for their relevant activity against Staphylococcus aureus and Gram-negative bacteria [64,78]. Moreover, some Pd(II) complexes reveal a higher antimicrobial activity compared with their free respective biologically active ligands [79,80]. Coliform bacteria are commonly used as indicators of the hygienic conditions of the home and work environments, and of food and water. The antibacterial response of the Pd1 and Pd2 was tested against the most common polluting bacterium of aqueous media, that is the ubiquitous *Escherichia coli* bacterium [81].

The Pd1 shows significant antibacterial activity. Following the standard UNI EN ISO 8199:2018 and the standard UNI EN ISO 9308-1:2017, the colony-forming units (CFU) were evaluated at different concentrations of Pd1 stock solutions (see Section 3.5). The sample obtained from the 0.1 mM stock solution, added to 50 mL of specific agar medium, showed 10 CFU/100 mL with respect to the standard reference (20 CFU/100 mL), i.e., about a 50% inhibition of the CFU. A greater CFU abatement was not found due to the precipitation of the complex in the more concentrated aqueous solution.

The Pd2 also shows significant antibacterial activity against the *Escherichia coli*, albeit lower than that of the Pd1. Nonetheless, its greater solubility in the aqueous medium guarantees a greater CFU abatement due to the more concentrated solutions. Specifically, following the standard UNI EN ISO 8199:2018 and the standard UNI EN ISO 9308-1:2017, the colony-forming units (CFU) were evaluated at different concentrations of the Pd2 stock solutions (see Section 3.5). The sample obtained from the 0.6 mM stock solution, added to 50 mL of the specific agar medium, showed 10 CFU/100 mL with respect to the standard reference (20 CFU/100 mL), i.e., about a 50% inhibition of the CFU. A greater CFU abatement (70%) was found by treating 50 mL of the specific agar medium with a 1.0 mM stock solution of the Pd2. A further abatement cannot be evaluated due to the precipitation of the complex in the more concentrated aqueous solution.

As expected, all the ligands showed lower antimicrobial activity when compared with their metal complex counterparts. The activity of the unbonded ligands were checked by the same UNI EN ISO 8199:2018 and UNI EN ISO 9308-1:2017 standard methods. For the compounds L and L1, a 50% inhibition of the CFU was obtained with the 15- and 20-mM stock solutions. A greater CFU abatement was not explored due to the precipitation of the compounds in the concentrated aqueous solution. The compound L2 produced a 50% inhibition of the CFU in the 10 mM solution. In this case, a greater CFU abatement (about 70%) was found with about 33 mM of the stock solution.

## 3. Experimental Section

### 3.1. Materials and Methods

The commercially available starting materials were supplied by Sigma Aldrich. The compounds L1 [20] and (PdL)_2_ [73] were obtained as previously described. The ^1^H NMR spectra were recorded in DMSO-d6 with a Bruker Advance II 400 MHz apparatus (Bruker Corporation, Billerica, MA, USA). The optical observations of the phase transitions were performed by using a Zeiss Axioscop polarizing microscope (Carl Zeiss, Oberkochen, Germany) equipped with an FP90 Mettler microfurnace (Mettler-Toledo International INC MTD, Columbus, OH, USA). The decomposition temperatures (5 wt.% weight loss) and the phase transition temperatures and enthalpies were measured under nitrogen flow by the employ of a DSC/TGA PerkinElmer TGA 4000 (PerkinElmer, Inc., Waltham, MA, USA), with a scanning rate of 10 °C/min. The absorption spectra were recorded by a JASCO F-530 spectrometer (scan rate 200 nm/min, JASCO Inc., Easton, MD, USA). The mass spectrometry measurements were performed using a Q-TOF premier instrument (Waters, Milford, MA, USA) with an electrospray ion source and a hybrid quadrupole time-of-flight analyser. The FT-IR spectra were recorded in KBr by a Shimadzu IRAffinity-1S apparatus.

### 3.2. Synthesis of L2

An amount of 0.286 g (1.00 mmol) of 4-((4-(tert-butoxy)phenyl)diazenyl)benzene-1,3-diol was dissolved at 70 °C in 20 mL of tetrahydrofuran 0.413 g (1.20 mmol) of 5-bromo-*N,N,N*-trimethylpentan-1-aminium bromide and mixed with 0.300 mg of K_2_CO_3_, added under stirring. After 1 h at the boiling temperature, the mixture was filtered and concentred under vacuum to about 5 mL. When the solution was cooled, the crude product precipitated in light yellow crystals. The compound was recovered and recrystallised from diethyl ether, with a yield of 65%, T_m_ = 185 °C (with decomposition). ^1^H-NMR (400 MHz, DMSO-d6, 25 °C, ppm): 1.03 (s, 9H), 1.48 (m, 2H), 1.78 (m, 2H), 1.87 (m, 2H), 3.08 (s, 9H), 3.73 (s, 2H), 4.11 (t, 2H), 4.21 (t, 2H), 6.64 (d, 1H), 6.77 (d, 1H), 7.12 (d, 2H), 7.59 (d, 1H), 7.77 (d, 2H).

^13^C-NMR (400 MHz, DMSO-d6, 25 °C, ppm): δ = 163.1, 162.0, 152.0, 144.3, 125.8, 123.0, 117.0, 115.2, 107.2, 104.3, 80.1, 68.6, 66.8, 54.9, 29.3, 27.2, 26.0, 25.1, 24.8 ppm. The elemental analysis calculated (%) for C_25_H_38_N_3_O_3_Br: C, 59.05; H, 7.53; N, 8.26; found: C, 59.22; H, 7.43; N, 8.16. HRMS(ESI): and the *m*/*z* calculated for C_25_H_38_N_3_O_3_^+^: 428.29; found 428.24 [M]^+^.

### 3.3. Synthesis of Pd1 and Pd2

The synthesis of the complexes Pd1 and Pd2 were performed from stoichiometric amounts of the L1 or L2, respectively, and the (PdL)_2_ by the same general procedure. As an example, in a typical preparation of Pd1, to 0.286 g (1.00 mmol) of L1 dissolved in 20 mL of dimethylformamide are added 0.150 g of potassium carbonate and, finally, 0.437 g of (PdL)_2_ (0.50 mmol). The colour turned from orange to red. After stirring about 1 h at 40 °C, the mixture was filtered and added to water, with precipitation ensuing. The red precipitate was recovered by filtration and crystallized in acetone, with a yield of 58%.

For the Pd2, the reaction solution was concentred under vacuum to about 5 mL, precipitated by adding cold water, recovered by filtration, and crystallized in acetone/hexane, obtaining an orange powder solid, with a yield of 45%.

For the Pd1: ^1^H-NMR (400 MHz, DMSO-d6, 25 °C, ppm) δ: 1.02 (s, 9H); 2.65 (s, 6H); 3.67 (s, 2H); 5.39 (d, 1H), 5.90 (d, 1H); 6.22 (d, 1H); 6.31 (d, 1H); 6.99 (d, 2H); 7.30 (d, 1H); 7.37 (d, 1H); 7.69 (d, 2H); 7.91 (d, 2H), 8.30 (d, 2H); 8.33 (s, 1H), 10.20 (s, 1H). See the hard copy in Appendix A (reported as a comparison with ligand L1).

^13^C-NMR (400 MHz, DMSO-d6, 25 °C, ppm): δ = 163.0, 161.8, 161.0, 154.0, 151.8, 151.4, 145.1, 144.0, 130.0, 126.7, 126.0, 125.3, 124.3, 123.0, 117.1, 115.0, 114.5, 113.0, 107.9, 104.23, 78.2, 38.0, 32.2, 26.8 ppm.

The elemental analysis calculated (%) for C_32_H_33_N_5_O_5_Pd: C, 57.02; H, 4.93; N, 10.39, Pd 15.79; found: C, 57.12; H, 4.98; N, 10.34; Pd 15.81. T_d_= 268 °C. HRMS(ESI): and the *m*/*z* calculated for C_32_H_34_N_5_O_5_Pd^+^: 674.16; found 674.20 [M + H]^+^.

The IR spectrum of the Pd1 compared with the L1 is reported in Appendix A.

For the Pd2: ^1^H-NMR (400 MHz, DMSO-d6, 25 °C, ppm) δ: 1.03 (s, 9H); 1.45 (m, 2H); 1.72 (m, 2H); 1.86 (m, 2H); 2.57 (s, 6H); 3.09 (s, 9H); 3.70 (s, 2H); 4.06 (t, 2H); 4.20 (t, 2H); 5.16 (s, 1H); 5.60 (d, 1H); 6.35 (d, 1H); 6.47 (d, 1H); 7.12 (d, 2H); 7.37 (d, 2H); 7.55 (d, 1H); 7.78 (d, 2H); 7.87 (d, 1H), 8.20 (d, 2H); 8.22 (s, 1H). See the hard copy in Appendix A (reported as a comparison with ligand L2).

^13^C-NMR (400 MHz, DMSO-d6, 25 °C, ppm): δ = 163.1, 161.8, 161.0, 154.0, 151.8, 151.5, 145.1, 144.0, 131.0, 130.8, 126.7, 126.0, 125.3, 124.3, 123.0, 117.1, 115.0, 114.5, 113.0, 107.9, 104.23, 80.1, 68.6, 66.8, 54.9, 40.1, 29.3, 27.2, 26.0, 25.1, 24.8 ppm.

The elemental analysis calculated (%) for C_40_H_51_N_6_O_5_BrPd: C, 54.46; H, 5.83; N, 9.53, Pd 12.06; found: C, 54.33; H, 5.77; N, 9.57, Pd 12.02. T_d_ = 250 °C. HRMS(ESI): and the *m*/*z* calculated for C_40_H_51_N_6_O_5_Pd^+^: 801.30; found 801.33 [M]^+^.

The IR spectrum of the Pd2 compared with the L2 is reported in Appendix A.

### 3.4. Single-Crystal X-ray Analysis

Single crystals of the Pd1 complex were prepared at room temperature by the slow evaporation of the complex in ethanol. The Pd1 crystals appeared as elongated plates, with typical dimensions of 0.7 × 0.3 × 0.3 mm. The data were collected with synchrotron radiation (wavelength, 0.7000 Å) from a XRD1 beamline at the ELETTRA Synchrotron Light Source, Trieste, Italy. By using a small loop of fine rayon fibre, the selected crystal for the data diffraction was dipped in cryoprotectant paratone oil and flash-frozen in a stream of nitrogen at 100 K. For the best diffracting crystals, a total of 360-degree crystal rotation data were collected from 720 images using an oscillation range of 0.5°. The data were processed using XDS and POINTLESS 1.11.21 with the data collection statistics reported in Appendix A [82,83]. The crystals had a monoclinic unit cell with axes a = 17.62 Å, b = 10.22 Å, c = 20.06 Å, b = 101.634°, V = 2917 Å^3,^ and space group *P* 2_1_/c. The structure solution was found by direct methods using SHELXS [84,85,86], which revealed the presence of one palladium ion in the ASU and most of the expected two ligands atom connectivity, corresponding to one Pd1 complex. The structure was anisotropically refined, using the full-matrix least-squares methods on the F^2^ against all the independent measured reflections, using SHELXL [87], run under the WinGX suite for the refinement of the small molecules [88]. A water molecule and an acetic acid molecule (neutral form) were found, mutually exclusive, with a partial occupancy of 0.4 and 0.6, respectively. The difference peaks on the electron-density map were observed corresponding to most of all the hydrogen atoms of the two ligands, which were introduced and refined in agreement with a riding model as implemented in SHELXL. The figures were generated using a Mercury CSD 3.6 [89]. The crystallographic data for the Pd1 have been deposited with the Cambridge Crystallographic Data Centre and can be obtained via www.ccdc.cam.ac.uk/data_request/cif (accessed on 24 July 2022).

### 3.5. Molecular Modelling

Ab initio simulations were performed adopting the Density Functional Theory (DFT) approach in the Linear Combination of Atomic Orbitals (LCAO) formalism. Each molecular orbit was expanded in a set of Gaussian Atomic Orbital (GAO) functions, centred in the nuclei position. The HOMO analysis was derived from the ground-state calculations, while the LUMO evaluation was performed considering the first excited state, adopting the Time-Dependent Density Functional Theory (TDDFT) formalism. The software adopted was ORCA 5.0.3 [90]. A global hybrid B3LYP functional [91], with a fraction of exact Hartree Fock exchange, was employed as previously described [92,93]. The Grimme empirical correction D3 was adopted for a consistent evaluation of the dispersion interactions [94,95] The large triple-ζ basis set is used, as the def-TZVPP of the Karlsruhe group [96]. This adopted basis set was previously adopted by us [97,98,99,100] as the best choice in terms of the computational performance and accuracy of the results, providing a reliable comparation with the experimental data.

### 3.6. In Vitro Antibacterial Activity Methods

Two stock solutions of the Pd1 and Pd2 were prepared by using 3% DMSO as the dissolving booster in ultrapure water. The Pd1 stock solution was 0.1 mM and the Pd2 stock solution was 1.0 mM. The official standard methods were employed by a UNI EN ISO-accredited laboratory. The standard UNI EN ISO 8199:2018 was followed for the preparation of the sample. The standard UNI EN ISO 9308-1:2017 was followed to evaluate the bacterial colonies. The standard specifies a method for enumerating the *Escherichia coli* and coliform bacteria based on the membrane filtration, the subsequent culture on a chromogenic agar medium for coliforms, and the calculation of the number of target organisms present in the sample. The sample preparation procedure and use of the methods for the water matrix agrees with UNI EN ISO 8199: 2018, and the formula used for the colony count was also always in accordance with the ISO. The sample of the wastewater was treated by filtering a water content of 100 mL on a nitrocellulose membrane with a porosity of 0.45 μm. The filtration system consists of a ramp with supports and containers (filter funnels) made of stainless steel, using as the suction system an electrically operated vacuum pump or a water pump. All the operations were performed in sterile conditions, the supports and containers being sterilised in an autoclave or being sterile disposable. The bacteria larger than the pores of the filter remain trapped on the surface of the filter itself. After the filtration, each membrane is placed in a plate containing a specific culture medium that allows the growth and differentiation of the bacteria sought. The used media were prepared in agreement with ISO 11133: 2018, respecting the conditions of selectivity, specificity, and sterility. In our case, the medium used for the *Escherichia coli* and for the total coliforms was chromogenic coliform agar. The plates, prepared with known concentrations of the complex being tested, were incubated at a temperature of 37 °C for 24 ± 2 h for both the investigated microorganisms. After the incubation time, the results were read by comparing the plates used as blanks, or without the complex, to those with the complex, observing a significant decrease in the number of colonies.

## 4. Conclusions

The cyclometallation of a Schiff base dye, followed by the bridge-splitting reaction with an azo chelating ligand, led to two stable and non-symmetric chromophore complexes, Pd1 and Pd2. Two *N,O* chelating diazo ligands with a different substituent and an antimicrobial potential were employed. The synthesized complexes are orange/red dyes with a relevant molar extinction coefficient, soluble in organic solvents, stable under daylight, ensuing a remarkable solvatochromism, depending on the solvent polarity. The structural features were acquired by a single crystal analysis of the Pd1. The electronic charge-transfer process was examined considering the differential electron density of the complex Pd1 compared with its ligands. Interestingly, the metal is involved in the charge-transfer process as an active bridge between the ligands. The antibacterial response of the two novel complexes was tested against *Escherichia coli*. Remarkably, the response of the two novel complexes revealed an increase in the antimicrobial activity by a ten- to hundred-fold factor with respect to the uncoordinated ligands. As expected, the azo ligand with a cationic chain increases the solubility of the complex Pd2 vs. the Pd1. A relevant 70% inhibition of the CFU against the *Escherichia coli* was obtained with a 1.0 mM aqueous solution of the Pd2 complex.

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
