# Peer review of "Insights into Two Novel Orthopalladated Chromophores with Antimicrobial Activity against *Escherichia coli"

_molecules, 2022, doi:10.3390/molecules27186060_

Round 1

Reviewer 1 Report

This submission by Rosita Diana and co-workers describes the syntheses, DFT calculations, and antimicrobial activity of two Pd(II) Schiff-base complexes. Authors used elemental analysis, HNMR, and single crystal X-ray diffraction for structural characterization of two complexes but put the biggest emphasis on its promising antimicrobial properties tested on Escherichia coli. The article draws some reasonable conclusions, has a good scientific basis and is structured well with sufficient experimental data. In my opinion, the manuscript nicely lies in scope of Molecules and is acceptable after major revision. Please consider the following some remarks which should be addressed by Authors:

1. The purity of the products should be tested by appropriate methods. It is well known that the purity of synthetic compounds can strongly affect the experimental results. 2. About antibacterial activity in vitro, the antibacterial activity of two complexes should be compared with that of the parent ligands, because some ligands have higher antibacterial activity than their corresponding complexes. Accordingly, in Abstract, "Some Pd(II) complexes exhibit a higher antimicrobial activity when compared to their uncoordinated biologically active ligands" may be rewritten.

3. Two complexes lack the complete structural characterization. The authors should supplement the FT-IR, MS, and 13C NMR data of two complexes in Experimental section. The FT-IR, MS and NMR spectra of two complexes should be submitted as a separate supplementary material. IR and NMR spectra of two complexes must be interpreted in comparison with the parent ligand data.

 4. In 2.1 section, authors state that both complexes show good solubility in ordinary organic solvents. However, the stability of the complexes should be studied in solvents by means of 1H-NMR or Uv-Vis spectra for at least a period of one week.

5. There are some mistakes in the manuscript and need to be revised.

6. The relevant contents should be concise.

Author Response

We thank the reviewers for the constructive comments on the manuscript.

As indicated below, we have checked all the general and specific comments provided by the Referees and have made necessary changes accordingly to their indications.

Sincerely,

Barbara Panunzi

REFEREE 1

This submission by Rosita Diana and co-workers describes the syntheses, DFT calculations, and antimicrobial activity of two Pd(II) Schiff-base complexes. Authors used elemental analysis, HNMR, and single crystal X-ray diffraction for structural characterization of two complexes but put the biggest emphasis on its promising antimicrobial properties tested on Escherichia coli. The article draws some reasonable conclusions, has a good scientific basis and is structured well with sufficient experimental data. In my opinion, the manuscript nicely lies in scope of Molecules and is acceptable after major revision. Please consider the following some remarks which should be addressed by Authors:

1.The purity of the products should be tested by appropriate methods. It is well known that the purity of synthetic compounds can strongly affect the experimental results. 

We thank the Referee for the observation. We added in the Manuscript and in the Supplementary Material the required analyses (see point 3).

  1. About antibacterial activity in vitro, the antibacterial activity of two complexes should be compared with that of the parent ligands, because some ligands have higher antibacterial activity than their corresponding complexes. Accordingly, in Abstract, "Some Pd(II) complexes exhibit a higher antimicrobial activity when compared to their uncoordinated biologically active ligands" may be rewritten.

We thank the Referee for the comment. In fact, this has allowed an improvement of the article.

According with his/her suggesting, we checked the antimicrobial activity of all the unbonded ligands (for two of them checked in other articles with different methods) by using the same ISO methods used for the complexes. We added the results in the manuscript. We also rewrote Abstract accordingly.

  1. Two complexes lack the complete structural characterization. The authors should supplement the FT-IR, MS, and 13C NMR data of two complexes in Experimental section. The FT-IR, MS and NMR spectra of two complexes should be submitted as a separate supplementary material. IR and NMR spectra of two complexes must be interpreted in comparison with the parent ligand data.

We added the required 13C NMR and MS of the complexes in the Experimental Section. In the Supplementary Material we added IR and NMR spectra comparison of ligands and complexes.

  1. In 2.1 section, authors state that both complexes show good solubility in ordinary organic solvents. However, the stability of the complexes should be studied in solvents by means of 1H-NMR or Uv-Vis spectra for at least a period of one week.

In Pf. 2.1 we stated: “The absorbance pattern of the solutions used in the solvatochromic test remain unaltered for months under daylight at room temperature. Similarly, the soaked filter papers do not fade under exposure at daylight for over 2 months”.

Accordingly with Referee’s observation, we added two novel experiments. We recorded   1H-NMR before and after about 10 days, and absorbance spectra of Pd1 and Pd2 in the solid state (absorbance spectra added in the Supplementary Materials, maxima added in Table 1) after about 2 months. The stability was proven, as we discussed in the manuscript.

  1. There are some mistakes in the manuscript and need to be revised.
  2. The relevant contents should be concise.

Thank you to the Referee. We did revise the whole manuscript and emend typos and other mistakes.

Reviewer 2 Report

Manuscript Molecules 1857961 by Rosita Diana et al. reports the antimicrobial activity against Escherichia coli of Pd(II) compounds containing Schiff bases. The paper is incomplete and cannot be published in present form, with the reasons described below.

A deep study characterization of the solid structures of the compounds should be reported (IR analysis and UV-Vis spectra in solid state)

For Pd2 the authors described the compound crystalline. They should obtain single crystals and performed X-ray analysis.

In the crystal description the authors should investigate the noncovalent interactions (H-bonds, pi-pi staking)

The authors present in past papers a family of these type ligands. Why they used only two? For publication in Molecules a deep study of these ligands on the Pd(II) ions should be investigated.

In the biological studies I did not see any comparison with the ligands. What is their biologic effect against the Escherichia coli? The part of the biological activity should be more elaborated, with new test again other bacteria.

Should the cytotoxic studies not also include a control with a compound of known cytotoxicity?

Author Response

We thank the reviewers for the constructive comments on the manuscript.

As indicated below, we have checked all the general and specific comments provided by the Referees and have made necessary changes accordingly to their indications.

Sincerely,

Barbara Panunzi

REFEREE 2

Manuscript Molecules 1857961 by Rosita Diana et al. reports the antimicrobial activity against Escherichia coli of Pd(II) compounds containing Schiff bases. The paper is incomplete and cannot be published in present form, with the reasons described below.

-A deep study characterization of the solid structures of the compounds should be reported (IR analysis and UV-Vis spectra in solid state).

We thank the Referee. As we did reply to Referee 1, we added in the Manuscript and in the Supplementary Material the required analyses.

-For Pd2 the authors described the compound crystalline. They should obtain single crystals and performed X-ray analysis.

We thank the Referee for the observation. Unfortunately, we cannot obtain good quality single crystals of Pd2, employable for the structural analysis. Based on our experience, we can assume a similar coordination pattern, and theoretical deductions valid for both complexes.

-In the crystal description the authors should investigate the noncovalent interactions (H-bonds, pi-pi staking)

The noncovalent interactions have been described in the text and Figure S2 added in the Supplementary Material.

-The authors present in past papers a family of these type ligands. Why they used only two? For publication in Molecules a deep study of these ligands on the Pd(II) ions should be investigated.

We thank the Referee for the observation. We are aware that this study can be extended to many other dyes based on the same coordination value. Obviously, we had to focus our attention on some compounds, based on a logic. Specifically, by varying only one functional group we could increase the solubility and establish differences between the absorbance and antimicrobial pattern.

-In the biological studies I did not see any comparison with the ligands. What is their biologic effect against the Escherichia coli? The part of the biological activity should be more elaborated, with new test again other bacteria.

We thank the Referee for the comment. Comparison of complexes with ligands has allowed an improvement of the article. Specifically, we checked the antimicrobial activity of the ligands (for two of them checked in other articles with different methods) by using the same ISO methods used for the complexes and added the related data in the Manuscript.

Here we used Escherichia coli as a ubiquitous bacterium for comparison between the complexes, and now between complexes and ligands.  A more elaborated bioactivity with test again other bacteria could be the subject of further work.

-Should the cytotoxic studies not also include a control with a compound of known cytotoxicity?

Our antimicrobial tests were performed by an accredited Laboratory by employ standard internationally validated ISO methods, carried out respect to the standard reference according to the protocol, as described in the text.

Round 2

Reviewer 1 Report

Accept

Reviewer 2 Report

The paper can be published în present form